# Self-Curing Glass Ionomer Cement Covered by Photopolymerizable Adhesive for Protection of Mucoperiosteal or Gingival Flap Sutures in Canine Oral Surgery

**DOI:** 10.3390/ani13162648

**Published:** 2023-08-17

**Authors:** Salviano Tramontin Bellettini, Regiane Pereira Baptista da Silva, Diogo Fernandes Giovanelli, Emerson Luiz Botelho Lourenço, Elton da Cruz Alves Pereira, Karina Sakumoto, Daniela Dib Gonçalves, José Ricardo Pachaly

**Affiliations:** 1Graduate Program in Animal Science with Emphasis on Bioactive Products, Universidade Paranaense (UNIPAR), Umuarama 87502-210, Brazil; regiane.baptista@edu.unipar.br (R.P.B.d.S.); dfgiovanelli@gmail.com (D.F.G.); 2Graduate Program in Biotechnology Applied to Agriculture, Universidade Paranaense (UNIPAR), Umuarama 87502-210, Brazil; emerson@prof.unipar.br (E.L.B.L.); dr.eltoncruz@gmail.com (E.d.C.A.P.); 3Graduate Program in Medicinal and Phytotherapeutic Plants in Primary Care, Universidade Paranaense (UNIPAR), Umuarama 87502-210, Brazil; karina.sakumoto@edu.unipar.br (K.S.); danieladib@prof.unipar.br (D.D.G.); 4Brazilian Institute of Specialties in Veterinary Medicine (ESPECIALVET), Maringá 87014-080, Brazil; pachaly@uol.com.br

**Keywords:** veterinary dentistry, veterinary oral surgery, dental extraction, oronasal fistula, dog

## Abstract

**Simple Summary:**

This study evaluated the efficiency of self-curing glass ionomer cement, covered by photopolymerizable adhesive, as a protective element for mucoperiosteal or gingival flap sutures, as investigated in oral surgery in 15 dogs, which needed oral surgery to correct oronasal fistulae, defects or oral cavity communications. G1—seven animals presenting oronasal fistulae after extraction of maxillary canine teeth, reduced by double-mucoperiosteal-flap technique. G2—five animals presenting oronasal fistulae after extraction of maxillary canine teeth, reduced by single-flap technique. G3—three animals presenting oronasal fistulae, two after maxillary fracture and one after excision of neoplasia. We used simple interrupted sutures with 3.0 nylon, and a thin layer of self-curing glass ionomer cement was applied immediately over the operated area. After cement’s settling time, a thin layer of photopolymerizable adhesive was applied to the already polymerized cement. The protective cement was removed after 6 to 15 days, and the result was healing of 100% of the oronasal fistulae. We concluded that this technique provides protection of sutures and surgical wounds, showing potential for routine use in oral surgery in dogs.

**Abstract:**

Periodontal disease is one of the main affections of the oral cavity of dogs. Its main complication is the formation of periapical abscess, which, when affecting the maxillary canine teeth, can lead to the formation of oronasal fistulae. The objective of this study was to evaluate the efficiency of self-curing glass ionomer cement, covered by photopolymerizable adhesive, as a protective element for mucoperiosteal or gingival flap sutures in oral surgery of dogs. We studied 15 dogs from the clinical routine of the dental service of a teaching veterinary hospital, which needed oral surgeries to correct oronasal fistulae, defects or oral cavity communications, regardless of the causal agent. Group one (G1) was composed of seven animals that presented oronasal fistulae after the extraction of maxillary canine teeth compromised by severe periodontal disease. These fistulae were reduced by the double-mucoperiosteal-flap technique, 15 days after the dental extraction. Group two (G2) was composed of five other dogs that presented oronasal fistulae after the extraction of maxillary canine teeth compromised by severe periodontal disease. In this group, the fistulae were reduced by the single-flap technique, immediately after the dental extraction. Group three (G3) was composed of three animals, two of which presented oronasal fistulae due to maxillary fracture and the third one after excision of oral neoplasia. In all groups, simple interrupted sutures were used with 3.0 nylon, and a thin layer of self-curing glass ionomer cement was applied immediately over the operated area. After cement’s settling time, a thin layer of photopolymerizable adhesive was applied to the already polymerized cement. In G1, the protective cement was removed on average at 15 (±2) postoperative days, in G2 at 6 (±1) days and in G3 at 11 (±9) days. In the postoperative period, the animals received antibiotics and anti-inflammatory drugs, and they received their usual dry dog food diet. No Elizabethan collar or any other protective measure was used for suturing or the surgical wound. The result was healing of 100% of the oronasal fistulae, without suture dehiscence or the need for new surgical interventions. Thus, it was concluded that the use of self-curing glass ionomer cement covered by photopolymerizable adhesive was fully satisfactory, providing protection of sutures and surgical wounds and showing the potential for routine use in oral surgery in dogs.

## 1. Introduction

Worldwide, veterinary dentistry has shown rapid evolution and growing sophistication, in order to better the quality of life for dogs and cats [1], in which periodontal disease is the most commonly diagnosed disease [2]. In these animals, the occurrence of oral diseases can reach more than 85% of the population over six years of age [3]. In addition to the oral cavity itself, the veterinarian should also be concerned about the implications of oral health for other organ systems. Periodontal disease is the inflammation caused by bacterial plaque buildup in the teeth [3,4], which is responsible for destroying the periodontal structures that support and protect the tooth (gum, alveolar bone, cementum and periodontal ligament) [3,5,6,7,8].

Periodontal disease is the most common cause of tooth loss in dogs [4] and may be associated with systemic diseases [9,10]. One of its common consequences is the formation of periapical abscesses due to the migration of bacteria along the periodontium to the root apex [3,9].

Currently, there are several therapeutic options for periodontal disease, including home care, professional dental cleaning, periodontal surgery and extraction, and all involve control and removal of bacterial plaque [11].

Often, oronasal fistulae do not cause clinical signs and it is difficult to detect small lesions during routine physical examination of restless patients, so anesthesia may be required [5,6,12,13,14,15].

The complications of flap techniques in dogs are many and surgical success depends on several factors that can prevent healing, thus defining a poor prognosis, such as dehiscence due to high suture tension, lack of blood supply, infection, lack of support for the flap, trauma in the preparation of the flap, tongue movement over the repair and inadequate postoperative care [5,14,16,17,18]. Studies in people indicate that relapses can reach 27% [19] and 34% [20].

Postoperative care should include antibiotic therapy, cleaning of the oral cavity with chlorhexidine antiseptic solution, offering light food for 14 to 21 days and preventing the animal from chewing on hard objects [13,21].

Glass ionomer is the generic name of a group of materials based on the reaction between silicate glass powder and polyacrylic acid, the name being due to the formulation, which comprises a glass powder and an ionomer containing carboxylic acids. The cement was originally designed for aesthetic restorations of human anterior teeth with class III and class IV cavity preparations. Due to its adhesion to the dental structure and potential to prevent tooth decay, the use of glass ionomer has increased, with it being indicated as cementation agents, orthodontic bracket bonding adhesives, groove and fissure sealers, linings, bases, core buildups and intermediate restorations [22].

The main properties of glass ionomer cement include chemical adhesion to enamel and dentin tissue; ability to release and incorporate or recharge fluorides; modulus of elasticity similar to that of dentin; biocompatibility with dental pulp and gingival mucosa; possibility of maintaining marginal sealing for long periods; and coefficient of linear thermal expansion similar to that of the dental structure, allowing it to prevent bacterial infiltration at the interface between tooth and restoration [23]. Furthermore, glass ionomer cements release fluorides, which are bacteriostatic or bactericidal [22].

The objective of this study was to verify the efficiency of the use of self-curing glass ionomer cement covered by light-curing adhesive as a protective element of mucoperiosteal or gingival flap sutures in canine oral surgery, contributing to the healing process by protecting the surgical wound and thus reducing the surgical wound dehiscence index and the postoperative time.

The hypothesis is that this dressing will be used as a new auxiliary therapeutic alternative for suturing mucoperiosteal or gingival flaps in surgical processes in animals in the scope of veterinary dentistry, with the purpose of protecting surgical wounds and their sutures, thus reducing the rate of stitch dehiscence in oral surgeries.

## 2. Materials and Methods

Fifteen dogs, patients of the clinical routine of the Small Animal Dental Service of the Paranaense University Veterinary Hospital (Umuarama, Paraná, Brazil), were included in this study. Regardless of the causal agent, these dogs all underwent oral surgery to correct oronasal fistulae, defects or oral cavity communications. The research project was registered at the Paranaense University Institute of Research, Studies and Science (IPEAC/UNIPAR) and approved by the university’s Research Ethics Committee (CEPEEA/UNIPAR), with protocol number 31348.

For better data evaluation, the animals were divided into three groups based on the performed surgical treatment (G1, G2 and G3). G1 and G2 consisted of animals with severe periodontal disease that led to an oronasal fistulae, the difference being the surgical technique used (double or single flap, respectively). G3 was composed of animals whose causative agent of oronasal fistulae was not periodontal disease.

In G1, the animals presented oronasal fistulae resulting from the extraction of maxillary canine teeth as a result of severe periodontal disease (Figure 1), and their fistulae were treated using the double-flap technique, performed on average 15 days after the periodontal and extraction procedure that caused the defect. Following the literature [17,21], the double-flap technique was chosen because the fistulae were large, with significant bone loss, defined edges and visible communication between the oral and nasal cavities.

In G2, the animals also presented oronasal fistulae resulting from the extraction of maxillary canine teeth due to the consequences of severe periodontal disease (Figure 2). However, their fistulae were treated using the single-flap technique, applied immediately after the periodontal and extraction procedure that caused the defect. This technique was due the small size of the fistulae [17].

In G3, the first dog presented an unconsolidated maxillary fracture with unilateral oronasal fistula for six months, and after removing the bone fragment containing the 101, 102, 103, 104, 105 and 106 teeth, the fistula was treated immediately using the single-flap technique (Figure 3). The second animal presented complete left maxillary fracture and bilateral oronasal fistulae as consequences of being bitten by another dog. After surgical treatment of the fracture, which resulted in the extraction of the 101, 102 and 103 teeth, the hard palate was sutured with simple interrupted sutures with 3.0 nylon, covering the bone tissue and closing the oronasal fistulae (Figure 4 and Figure 5). Finally, the third G3 patient presented a small oronasal fistula resulting from the removal of a neoplastic mass located in the region of the 203 tooth, which was treated using the single-flap technique immediately after the extraction of that tooth (Figure 6).

Prior to surgical procedures, a complete blood count and routine blood chemistry test were performed. All animals underwent dissociative anesthesia by intramuscular administration of the combination named “ZAX” (tiletamine (Zoletil™, Virbac, São Paulo, SP, Brazil), zolazepam (Zoletil™, Virbac, São Paulo, SP, Brazil), xylazine (Rompun™, Bayer, São Paulo, SP, Brazil) and atropine (Atropivet™, Uzinas Chimicas Brasileiras S/A, Jaboticabal, SP, Brazil), previously combined in the same vial) [24], in doses calculated by allometric scaling [25]. In all 15 cases, the patients did not undergo endotracheal intubation, instead breathing atmospheric air during the surgical procedures. All liquids produced during the surgeries were removed by dental wet aspiration equipment and surgical towels.

In the preoperative period, immediately after anesthesia, all patients received intramuscular amoxicillin (Bactrosina™, Bayer, São Paulo, SP, Brazil) (30.0 mg/kg) and subcutaneous meloxicam (Maxicam™, Ourofino, Cravinhos, SP, Brazil) (0.2 mg/kg, SID). The surgical procedure was preceded by oral cavity washing, cleaning and antisepsis with 0.12% chlorhexedine digluconate solution [12,15]. During the surgical procedures for closing the oronasal communications, regardless of the surgical technique employed (single or double flap), the simple interrupted suture pattern with 3.0 nylon (Nylon 3-0, Shalon Suturas, São Luiz de Montes Belos, GO, Brazil) was used. After suture was applied, the oral cavity was cleaned again in the same manner and completely dried using gauze pads and compressed air.

Subsequently, a thin layer of self-curing glass ionomer cement (Figure 7) was applied to the surgical suture with a brush, allowing the material to set for eight minutes, until it polymerized over the wound and suture.

After the glass ionomer cement’s self-curing, a thin layer of light-curing adhesive (Adper Single Bond 2™, 3M ESPE, Sumaré, SP, Brazil) was also brushed over the self-curing glass ionomer layer to make it more solid and moisture-resistant, thereby protecting it from aggression from the oral cavity environment (Figure 8).

After application, the adhesive was cured using a portable dental curing light unit (Ultralumen SL™, Sanders, Santa Rita do Sapucaí, MG, Brazil), in five sessions of 10 s each, as recommended by the manufacturer (Figure 9). Thus, a protective dressing was obtained, covering the surgical wound and sutures, adhering to the suture knot cords in a single block.

In the postoperative period, starting after the surgery, all dogs received amoxicillin (Amoxil™, GlaxoSmithKline Brasil Ltd.a, Rio de Janeiro, RJ, Brazil) (30.0 mg/kg, BID, for seven days) and meloxicam (Maxicam™, Ourofino, Cravinhos, SP, Brazil) (0.1 mg/kg, SID, for three days). All patients were discharged from the hospital on the same day as surgery, after recovering from anesthesia. Normal feeding with dry commercial dog food was recommended, and no collar or any other protective measure or device was indicated.

The macroscopic aspects of the surgical wounds were evaluated every two or three days until the self-curing glass ionomer cement covered with light-cured adhesive was removed and we observed complete healing of the surgical wound. Signs of inflammation, healing and suture dehiscence (stitch loss and discontinuity in the healing line) were inspected, as well as the permanence of the dressing in place. The surgical sites were photographed in all evaluations. When chemical restraint was needed for these procedures, the animals underwent the same anesthetic protocol used during the surgery. All the animals were reassessed every 30 days over 8 months to exclude recurrences.

## 3. Results

All patients were within normal laboratorial standards for dogs.

G1 grouped seven animals, four females and three males, weighing between 3.2 and 8.0 (5.1 ± 1.5) kg and aged between 6 and 14 (11 ± 3) years old. These dogs were a Brazilian Terrier, three Poodles and three mongrel dogs.

In the seven animals of G1, the removal of the cement protection occurred between 12 and 19 (15 ± 2) days after performing the surgical procedure to correct the fistulae, except in one dog. In that patient, the right upper canine (104) was extracted, and the protective dressing was removed on the sixth postoperative day, when a completely healed surgical wound was observed, with persistence of a single suture causing a small area of inflammation (Figure 10). Three days after removal of this suture, the palatal mucosa was completely healed without signs of inflammation. In the other six G1 animals, three fistulae were due to the extraction of the left upper canine (204), two the extraction of the upper right canine (104), and one the extraction of both the right upper canine (104) and the left upper canine (204). The patient who presented two oronasal communications had both operated on at the same surgical time. In G1, in general terms, the removal of the protective dressing was performed when it was detached from the gingival and palatal mucosa, forming a block that was attached only to the sutures (Figure 11 and Figure 12). All G1 patients presented complete healing of the surgical wound after removal of the self-curing glass ionomer cement covered by light-curing adhesive. All of them presented a small area of inflammation in the area located under the glass ionomer cement, and three days after the removal of the dressing encompassing the sutures and the cement, the mucosa was completely healed, with no signs of inflammation. G1 patients were monitored between 45 and 230 days postoperatively and none had recurrences or postoperative complications (Figure 13 and Figure 14).

G2 grouped five animals, two females and three males, weighing between 3.8 and 11.8 (7.2 ± 3.3) kg and aged between 4 and 13 (9 ± 4) years old. These dogs were one Miniature Schnauzer, one Miniature Pinscher, one mongrel dog and two Poodles.

In G2, the protective dressing was removed between the fifth and eighth postoperative day (6 ± 1), when the glass ionomer no longer adhered to the gingival and palatal mucosa, and was only adhered to the sutures in a single cohesive block (Figure 15). Complete healing was observed upon removal of dressing and sutures, with small areas of inflammation remaining (Figure 16), and three days after removal, there was complete healing without signs of inflammation. G2 patients were monitored for 60 days after removal of dressing and sutures, without any signs of complications related to the procedures.

G3 grouped three male dogs, weighing between 2.5 and 34.0 (14.4 ± 17.1) kg and aged between 2 and 10 (6 ± 2) years. These dogs were one Poodle, one Miniature Pinscher and one Labrador Retriever.

In the three G3 animals, the removal of the protective dressing occurred between 3 and 21 (11 ± 9) days after surgery. In the patient that presented unconsolidated maxillary fracture with unilateral oronasal fistula for six months before surgery, the protective dressing and sutures were removed on the third postoperative day. In that time, the self-curing glass ionomer cement covered by light-curing adhesive was no longer attached to the gingival and palatal mucosa, being attached only to the sutures (Figure 17). At the time of removal, the surgical wound was completely healed, presenting small areas of inflammation, and three days later, the oral mucosa was completely healed, with no signs of inflammation. In the patient presenting complete maxillary fracture and bilateral oronasal fistulae as consequences of being bitten by another dog, the protective dressing was removed on the 11th day after the defect repair surgery, when there was no adherence to the palatal and gingival mucosa, with adhesion only to the sutures, as a single cohesive block (Figure 18). After removal of the protective dressing, the surgical wound was completely healed, with no oronasal communication or bone exposure, presenting only small areas of inflammation (Figure 19). Ten days after removal, the patient was reassessed and its oral mucosa was completely healed, with no signs of inflammation (Figure 20). In the third patient of G3, who presented a small oronasal fistula resulting from the removal of a neoplastic mass (inflamed fibroepithelial polyp) located in the region of the 203 tooth, which was treated using the single-flap technique immediately after tooth extraction, the protective dressing was already absent 21 days after surgery. At that time, the gingival and palatine mucosae were completely healed, with no signs of inflammation or recurrence of the neoplastic mass. The animal was monitored for 12 months, with no late recurrence of the tumor or oronasal communication.

## 4. Discussion

Periodontal disease is a common oral health issue in dogs. Around 85% of dogs over the age of six years have some kind of periodontal disease. [2,3]. In this study, patients from G1 and G2 had mean ages of 11 and 9 years, respectively. The disease affects smaller and narrow-muzzled dogs [2,13,26], as observed in this study, in which animals from G1 and G2 had these physical characteristics. Periodontal disease stands as the primary reason for tooth loss in dogs [4]. In this study, dogs from G1 and G2 presented severe periodontal disease and underwent maxillary canine teeth extraction. The situation caused oronasal fistula and led to the need for surgical treatment, corroborating data presented by other authors [3,5,7,8,9,10,12,27], who stated that the oronasal fistula is usually secondary to severe periodontal diseases or maxillary canine extraction.

Periodontal treatment is important because the health of the oral cavity is fundamental to the overall well-being of the animal. It directly assists in the prevention of diseases, complications, pain and discomfort caused by the condition, contributing to the improvement of the quality of life. During this study, all procedures for the treatment of periodontal disease followed the directions of the World Small Animal Veterinary Association Global Dental Guidelines [28].

Two G3 patients presented maxillary fractures, which most common causes in dogs being traumatic, usually associated with being run over by cars. Other causes include fighting with other animals, falls, blunt injuries and gunshot wounds [29]. As complications associated with mandibular fractures, the same authors list dental trauma, malocclusion, oronasal fistulae and palatal defects, osteomyelitis and bone sequestration, nonunion, facial deformities and abnormal dental eruption. In this study, one animal presented unconsolidated maxillary fracture with unilateral oronasal fistula, corroborating data that mention oronasal fistulae, palatal defects and nonunion as consequences of mandibular fractures [29]. The second G3 patient presented a complete maxillary fracture and bilateral oronasal fistulae resulting from another dog’s bite, corroborating the citation about fights with other animals as a cause of maxillary fractures with oronasal fistulae as consequences [29]. Surgical complications originating from maxillary neoplasms have a high incidence in dogs, including infection, flap necrosis, suture dehiscence and occurrence of oronasal fistulae [30], confirming that observed in the third patient of G3, where oronasal fistula resulted from excision of a tumor mass located in the 203 tooth region.

As many dogs with oronasal fistulae are geriatric patients [6,21], laboratory tests prior to general anesthesia are indicated, as well as a chest radiograph to check for possible aspiration pneumonia [6,21]. In this study, laboratory screenings were performed for all animals, with all results within the limits recommended for the species. There was no indication for thoracic radiographic examination because no animal presented clinical signs of pneumonia.

Many authors [21,29] mention that general anesthesia with tracheal intubation is required for oral cavity procedures, so that a tube and cuff prevent penetration of blood and fluids into the lower airways. The same authors comment that tracheal intubation may hinder surgery to correct oronasal fistulae, and they suggest intubating through pharyngotomy or tracheotomy [21,29]. In this study, all animals underwent dissociative anesthesia by intramuscular administration of the combination named “ZAX” (tiletamine, zolazepam, xylazine and atropine, previously combined in the same vial) in allometrically scaled doses [25], and this protocol was efficient, allowing the accomplishment of all surgeries, without anesthetic intercurrences. The patients did not undergo endotracheal intubation, instead breathing atmospheric air during the surgical procedures, without any respiratory problem. All liquids produced during the surgeries were easily removed by dental wet aspiration equipment and surgical towels, without any problems related to these liquids. These data contradict the statements of the authors previously mentioned [21,29].

Pre- and postoperative antibiotic and anti-inflammatory therapy are a routine in canine dentistry [13,21], and the postoperative evolution of all cases occurred without infections.

For oral cavity washing, cleaning and antisepsis, some authors recommend diluting antiseptic solution in saline [20]. Many others indicate the use of 0.12% chlorhexidine digluconate [12,15], chosen for this study.

The simple interrupted suture pattern with 3.0 nylon used for surgical wound synthesis in this research differs from that proposed by other authors, who recommend the same interrupted simple pattern, but using absorbable monofilament suture [21,27]. It is noted that there is a disadvantage in the use of nonabsorbable sutures in veterinary patients due to the need for sedation or anesthesia to remove them [30,31]. In this study, however, the nylon suture was very efficient, since no animal presented dehiscence, and suture removal was performed under chemical restraint in all patients.

The complications of the flap techniques are many and the success of the surgical procedure depends on several factors, such as suture dehiscence due to high suture tension, lack of blood supply, infection, lack of support for the flap, traumatic surgery to prepare the flap, and especially tongue movement over the sutures [5,13,17,21,29,32]. 

In this study, the use of a self-curing glass ionomer cement layer covered by alight-curing adhesive layer applied over the sutured surgical wound acted as a protective dressing, preventing direct contact of the tongue with the operated area. This precluded suture dehiscence and avoided the need for new procedures in 100% of cases. There are no reliable data about recurrence in this kind of surgical procedure in dogs, but in people, recidivation occurs in 27% to 34% of cases [19,20].

The glass ionomer cement presents biocompatibility with the oral mucosa [23]; however, as observed in this work, contact with the oral mucosa led to inflammation, which completely regressed after removing the dressing. This paper presents the first report on the use of glass ionomer cement in the oral mucosa.

Throughout the postoperative period, the animals of the three groups were fed dry commercial food, contrary to what is recommended by many authors who indicate offering light food for two to three weeks and preventing the animal from chewing on hard objects, to avoid dehiscence of the sutures [13,21,29]. In this study, the self-curing glass ionomer covered with light-curing adhesive was effective as a protective dressing for the sutures and surgical wounds, allowing the animals to receive normal feeding, ensuring their well-being and optimizing home care.

The use of auricular cartilage in five oronasal fistulae cats resulted in the elimination of lesions [33], with an average healing time of six weeks, much higher than the average of 15 days observed in this study for animals operated on with the double-flap technique (between 12th and 19th day), and the average of six days when using the single-flap technique (between 5th and 8th day).

When working with experimentally induced oronasal fistula filled with acrylic resin in 12 dogs, total dehiscence was reported in 1 (8.3%), and partial dehiscence in 8 (66.6%) animals [34], while in the 15 cases described here, no recurrence was observed. In that study [34], new surgical procedures were necessary because the communication orifice remained occluded by the acrylic. The reported healing period was 10 to 21 days [32], similar to the 6 to 21 days observed in this study, but we treated natural occurring clinical cases of oronasal fistulae, rather than induced lesions.

The use of lyophilized inorganic bovine bone has been considered an effective method to repair artificially induced oronasal fistulae in healthy dogs [35]. Defects were filled in between 7 and 21 days postoperatively [35], which was similar to the period observed in the present study, when the double-flap technique was used. That research [35], however, reported a partial dehiscence index of 20%, requiring further intervention, compared to zero dehiscence registered in this report. As a notable difference, we treated naturally occurring fistulae in dogs as part of the routine of a small animal dentistry service, rather than induced lesions in healthy animals. Moreover, iatrogenic fistulae tend to have good healing and often do not require further interventions [27].

Although the results of this study are promising, the number of animals in the groups was still small. Further studies with a larger number of animals should follow to increase the casuistry and corroborate our results.

## 5. Conclusions

Based on the results of this study, it can be concluded that the use of self-curing glass ionomer cement covered with light-curing adhesive was fully satisfactory as a protective dressing for sutures and surgical wounds in the postoperative period of oronasal fistulae treatment in 15 dogs. Because the knot cords used in sutures are enclosed by cement in the form of a single cohesive block, it is assumed that there is a reduction in discomfort caused by constant tongue contact over exposed sutures. This new method demonstrates potential for routine use in canine oral surgery and should also be evaluated on other species.

## Figures and Tables

**Figure 1 animals-13-02648-f001:**
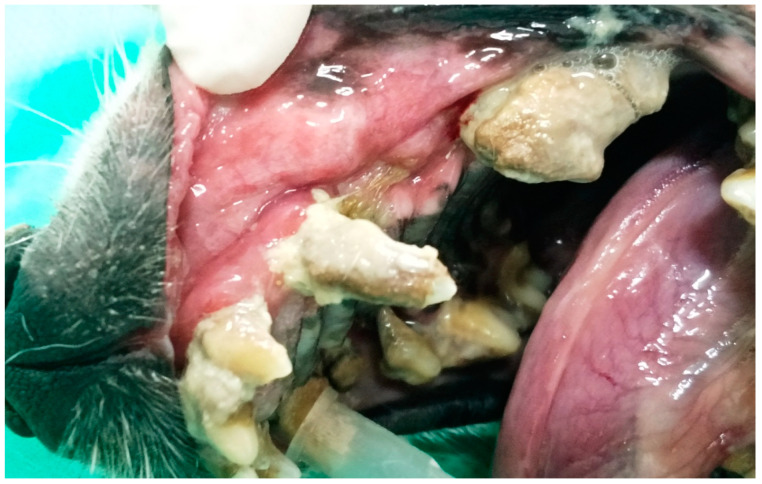
Case A. Image of the oral cavity of a 5.0 kg 13-year-old Brazilian Terrier dog with severe periodontal disease. Note gingival retraction, supragingival dental calculus and plaque buildup affecting the incisors, canine (with secondarily adhered hair), fourth premolar and first molar left maxillary teeth (this periodontitis caused an oronasal fistula after extraction of the left upper canine tooth).

**Figure 2 animals-13-02648-f002:**
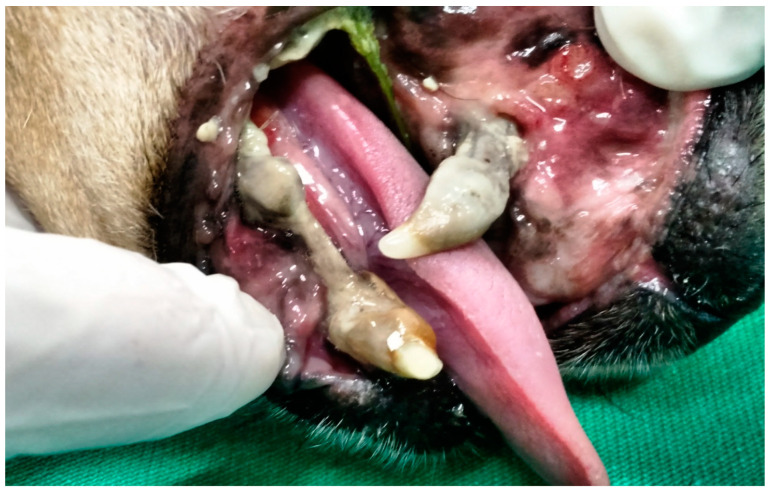
Case B. Image of the oral cavity of a 4.5 kg 10-year-old mongrel bitch with severe periodontal disease. Note gingival retraction, supra and subgingival plaque buildup, and dental calculus affecting the right upper canine, right lower canine and right lower premolar teeth (this periodontitis caused the oronasal fistula observed after extraction of the right upper canine tooth).

**Figure 3 animals-13-02648-f003:**
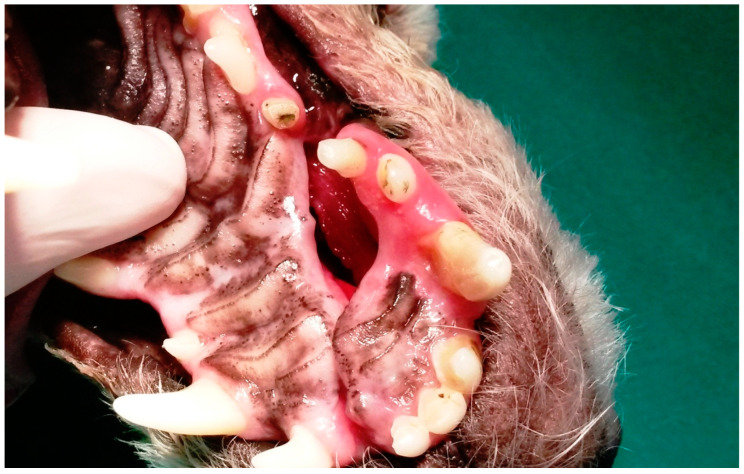
Case C. Image of the oral cavity of a 6.7 kg seven-year-old Poodle dog showing lesions that occurred six months earlier—fracture and bone nonunion in the right maxilla, with concomitant oronasal fistula.

**Figure 4 animals-13-02648-f004:**
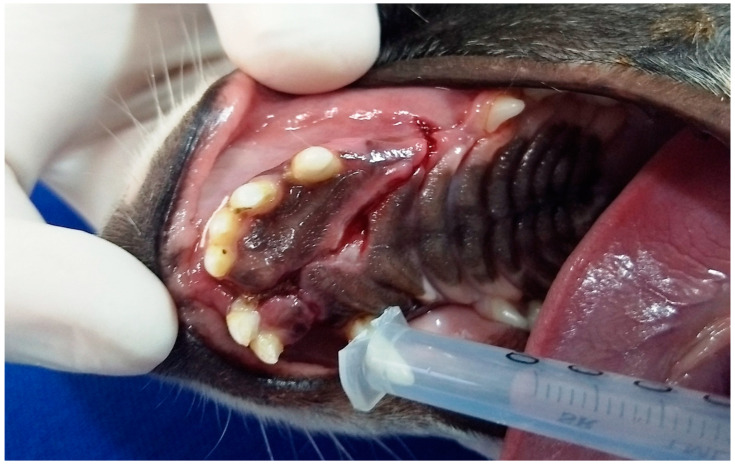
Case D. Image of the oral cavity of a 2.5 kg two-year-old Miniature Pinscher dog with left maxillary fracture and bilateral oronasal communication caused by a dog bite.

**Figure 5 animals-13-02648-f005:**
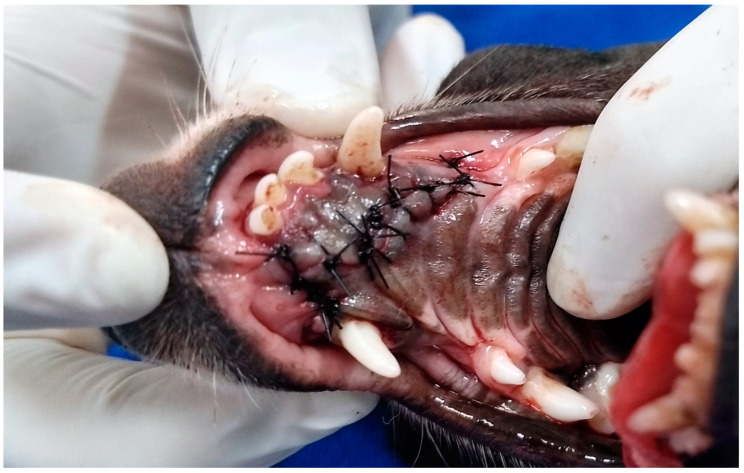
Case D. Image of the oral cavity of the same 2.5 kg two-year-old Miniature Pinscher dog presented in Figure 4, immediately after suturing the hard palate and gingival mucosa, covering the bone tissue and closing the bilateral oronasal communication, with simple interrupted suture with 3.0 nylon.

**Figure 6 animals-13-02648-f006:**
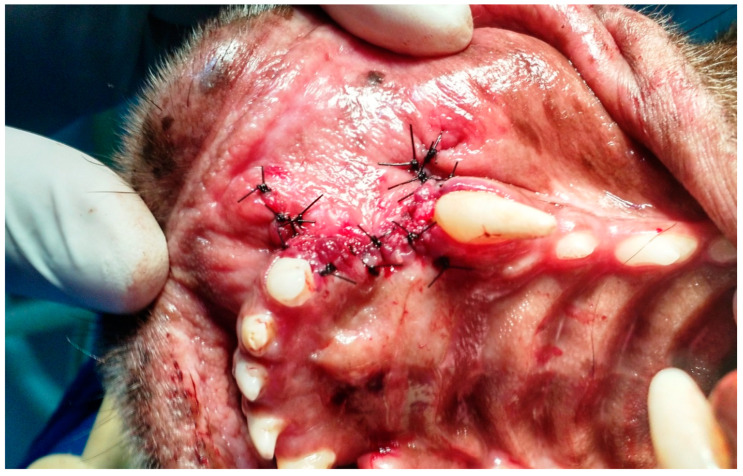
Case E. Image of the oral cavity of a 34 kg 10-year-old Labrador dog, immediately after the surgical procedure to close an oronasal communication caused by the excision of a neoplastic mass located in the region of the 203 tooth. Immediately after the tooth’s extraction, the fistula was closed by the simple flap technique. Note the simple interrupted suture with 3.0 nylon.

**Figure 7 animals-13-02648-f007:**
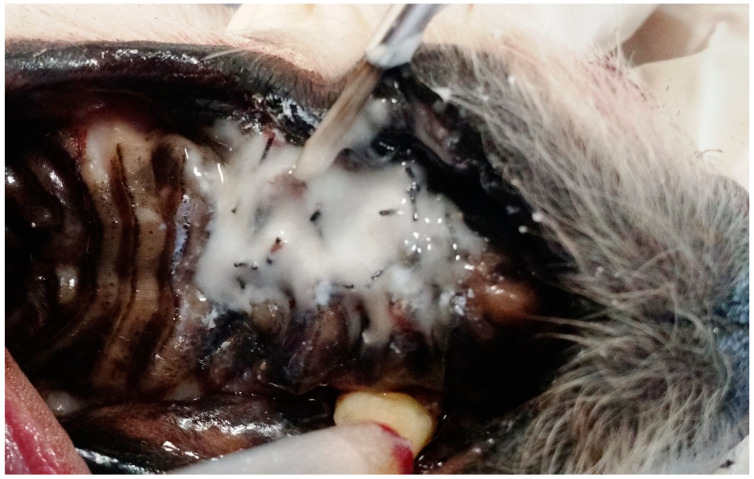
Case F. Image of the oral cavity of a 3.2 kg 14-year-old Poodle dog, immediately after closing of an oronasal communication due to the extraction of the right upper canine tooth, using the double-flap technique and simple interrupted suture with 3.0 nylon. Note the self-curing glass ionomer cement being applied on the surgical suture with a brush.

**Figure 8 animals-13-02648-f008:**
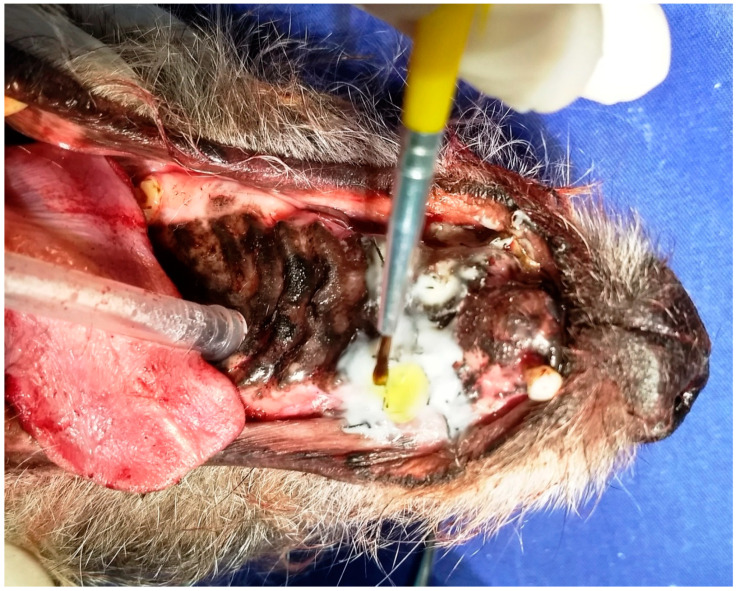
Case G. Image of the oral cavity of a 5.0 kg 10-year-old mongrel bitch after closing oronasal communications due to the extraction of the upper canine teeth, using the double-flap technique and simple interrupted suture with 3.0 nylon. Note the light-curing adhesive being applied with a brush over the previously applied and already cured glass ionomer cement.

**Figure 9 animals-13-02648-f009:**
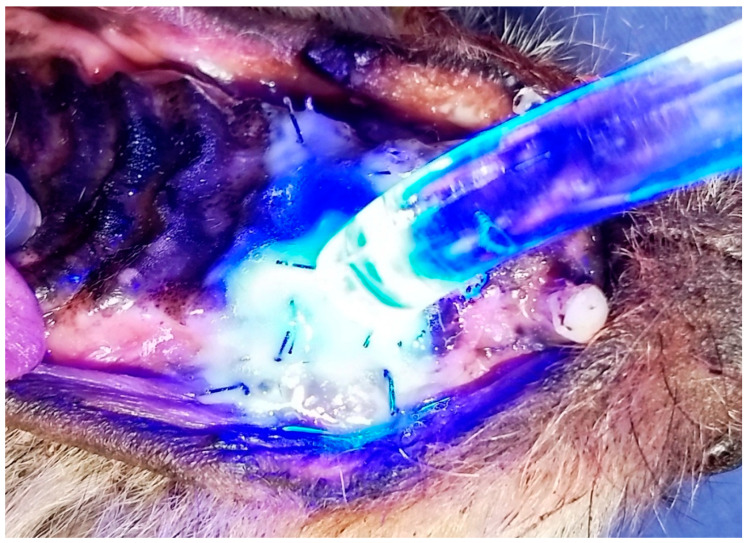
Case G. Image of the oral cavity of the same 5.0 kg 10-year-old mongrel bitch presented in Figure 8. Note the light-curing of the adhesive with a photopolymerizer.

**Figure 10 animals-13-02648-f010:**
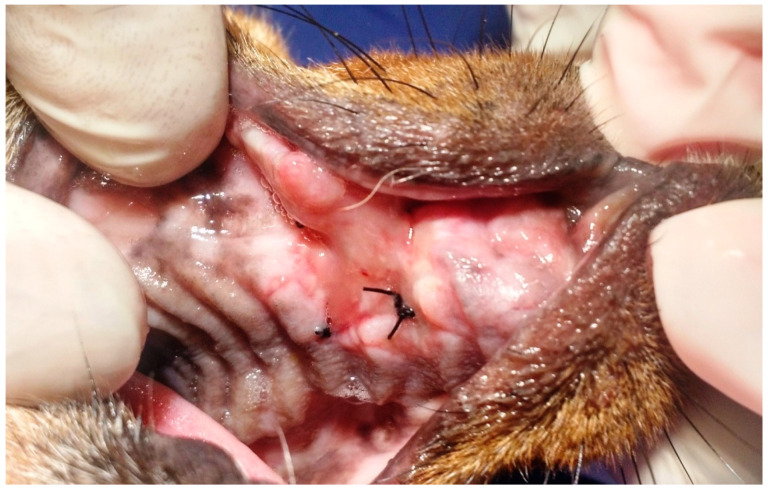
Case B. Image of the oral cavity of a 4.5 kg 10-year-old mongrel bitch, six days after applying self-curing glass ionomer cement covered by light-curing adhesive over the sutures used to close an oronasal communication due to the extraction of the upper right canine tooth. Note the absence of the protective dressing, healing of the surgical wound and persistence of some sutures.

**Figure 11 animals-13-02648-f011:**
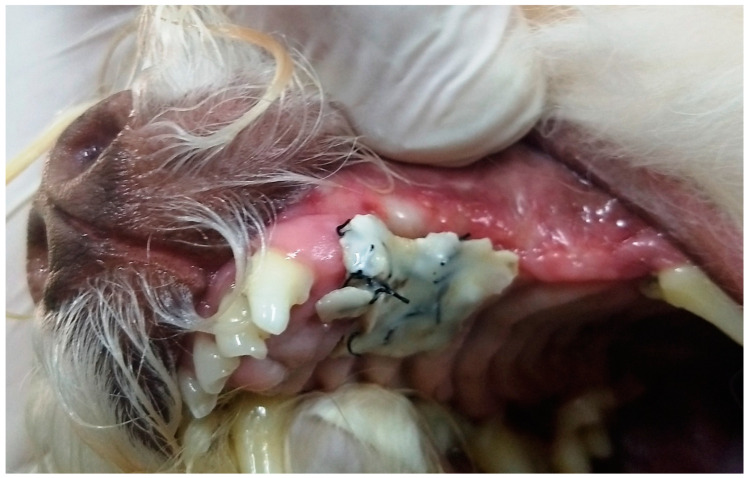
Case H. Image of the oral cavity of a 5.0 kg six-year-old Poodle dog 15 days after applying self-curing glass ionomer cement covered by light-curing adhesive over the sutures used to close an oronasal communication due to the extraction of the upper left canine tooth, using the double-flap technique and simple interrupted suture with 3.0 nylon. Note the protective dressing forming a cohesive block with the sutures, already cracked in some parts.

**Figure 12 animals-13-02648-f012:**
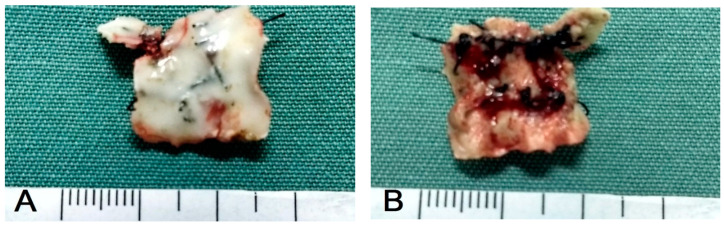
Case H. Appearance of the protective dressing immediately after being removed from the palate of the same 5.0 kg six-year-old Poodle dog presented in Figure 11. (**A**). Upper surface of the dressing seen as a cohesive block with the nylon sutures; (**B**). Lower surface, showing that the block even incorporated the suture knots.

**Figure 13 animals-13-02648-f013:**
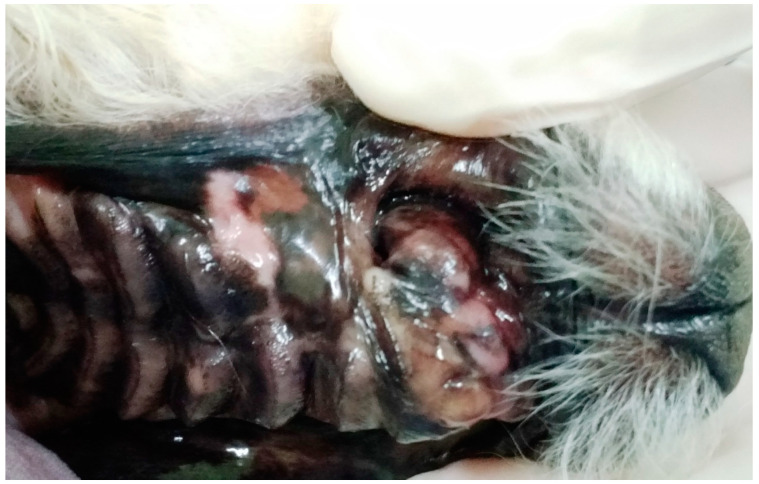
Case F. Oral cavity image of the same 3.2 kg 14-year-old Poodle dog presented in Figure 7, 99 days after closing of an oronasal communication due to the extraction of the right upper canine tooth. Note the fully recovered surgical area, including repigmentation of the hard palate.

**Figure 14 animals-13-02648-f014:**
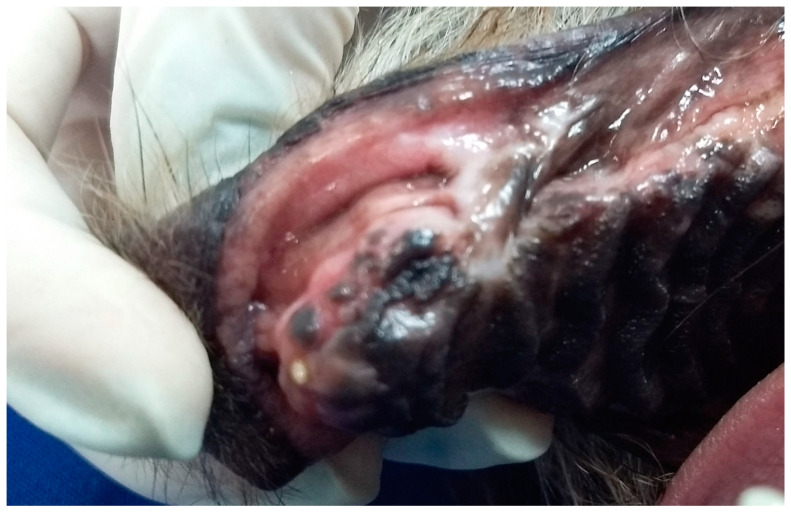
Case G. Image of the maxillary left side of the same 5.0 kg 10-year-old mongrel bitch presented in Figure 8 and Figure 10, 231 days after closing oronasal communications due to the extraction of the upper canine teeth. Note the complete healing of the surgical site.

**Figure 15 animals-13-02648-f015:**
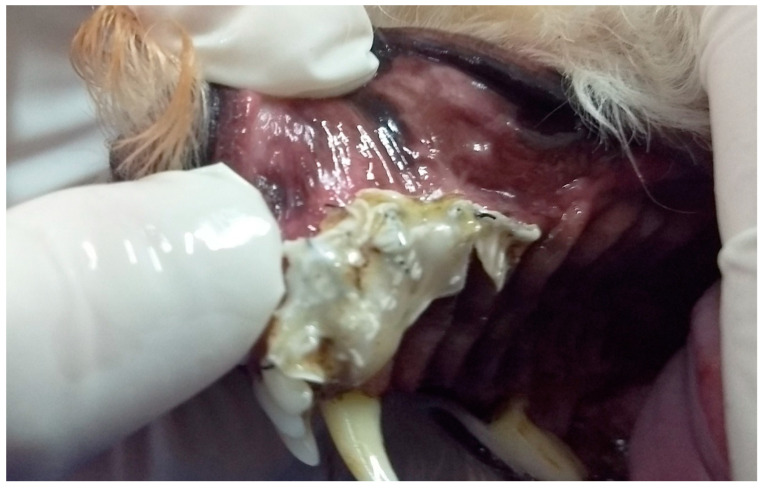
Case I. Image of the oral cavity of a 4.5 kg 13-year-old Poodle dog, eight days after applying self-curing glass ionomer cement covered by light-curing adhesive over the sutures used to close an oronasal communication due to the extraction of the upper left canine tooth, using the simple flap technique and simple interrupted suture with 3.0 nylon. Note the protective dressing forming a cohesive block with the sutures.

**Figure 16 animals-13-02648-f016:**
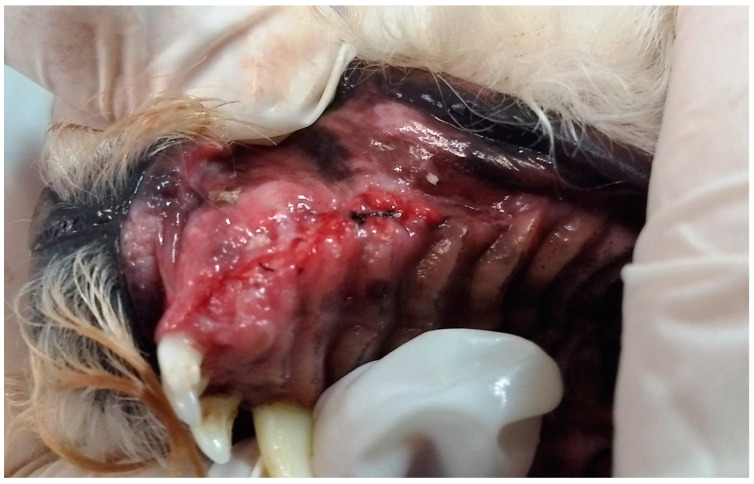
Case I. Image of the oral cavity of the same 4.5 kg 13-year-old Poodle dog presented in Figure 15, immediately after removal of the protective dressing. Note the complete healing of the surgical wound and some areas of inflammation.

**Figure 17 animals-13-02648-f017:**
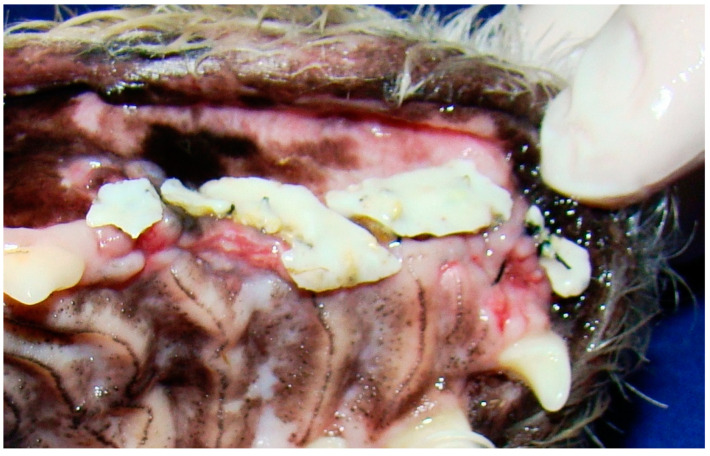
Case C. Image of the oral cavity of the same 4.5 kg 13-year-old 4.5 kg Poodle dog presented in Figure 3, three days after applying self-curing glass ionomer cement covered by light-curing adhesive over the sutures used to close an oronasal communication derived from fracture and bone nonunion in the right maxilla, with concomitant oronasal fistula. Note that the protect bandage is coming off the palate and has fragmented into four blocks that remain attached to the simple interrupted sutures of the simple flap used to close the defect.

**Figure 18 animals-13-02648-f018:**
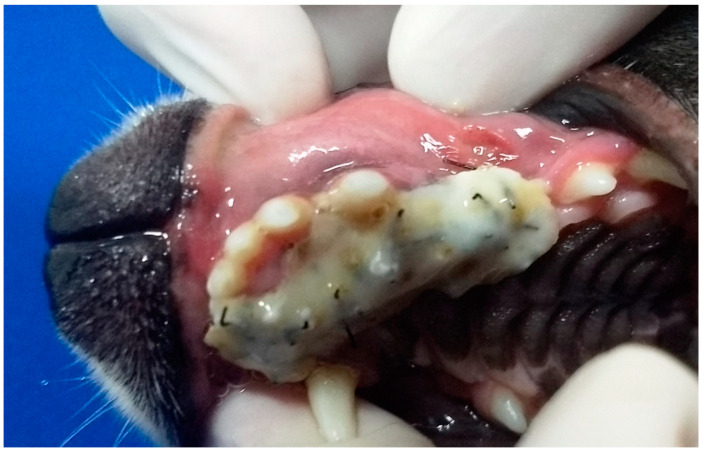
Case D. Image of the oral cavity of the same 2.5 kg two-year-old Miniature Pinscher dog presented in Figure 5 and Figure 6, 11 days after applying self-curing glass ionomer cement covered by light-curing adhesive over the simple interrupted sutures with 3.0 nylon used to close the oronasal communication due to left maxillary fracture caused by a dog bite. Note the protective dressing forming a cohesive block with the sutures.

**Figure 19 animals-13-02648-f019:**
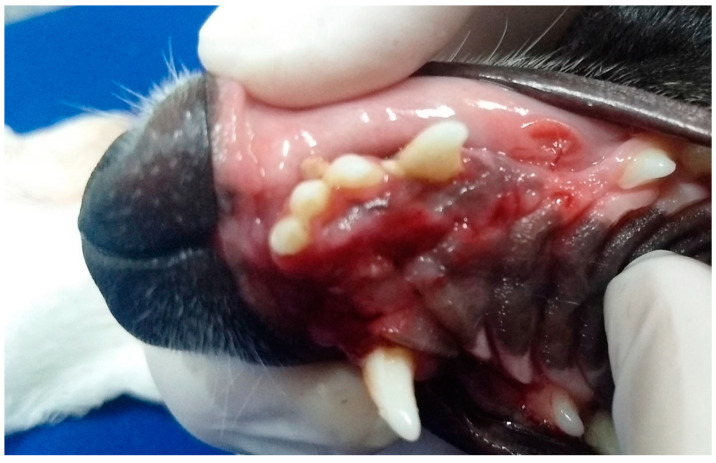
Case D. Image of the oral cavity of the same 2.5 kg two-year-old Miniature Pinscher dog presented in Figure 5, Figure 6 and Figure 19, 11 days after surgery and immediately after removing the protective dressing. Note the healing of the surgical wound and some areas of inflammation.

**Figure 20 animals-13-02648-f020:**
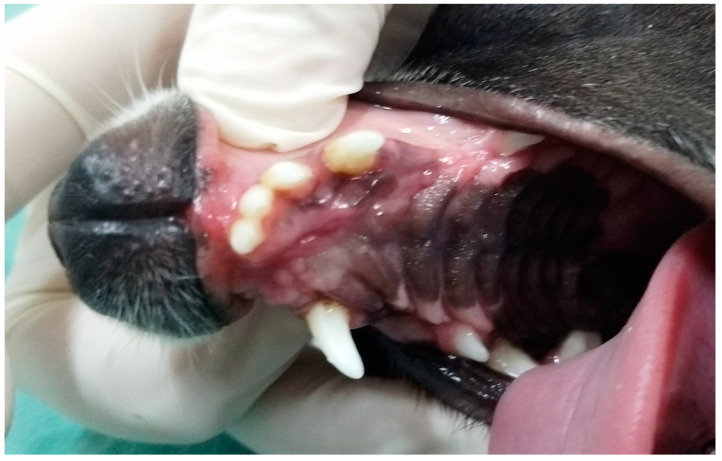
Case D. Image of the oral cavity of the same 2.5-year-old Miniature Pinscher dog, presented in Figure 5, Figure 6, Figure 19 and Figure 20, 21 days after surgery and 10 days after removing the protective dressing. Note complete healing of the surgical wound.

## Data Availability

The data presented in this study are available on request from the corresponding author. The data are not publicly available due to ethical and privacy issues.

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
