# Peer review of "Self-Curing Glass Ionomer Cement Covered by Photopolymerizable Adhesive for Protection of Mucoperiosteal or Gingival Flap Sutures in Canine Oral Surgery"

_animals, 2023, doi:10.3390/ani13162648_

Round 1

Reviewer 1 Report

This manuscript is well written and sets its objectives. However hypothesis is missing. 

Furthermore some of the English terminology is not used appropriatelly. For example the first sentence in introduction refers to domestic and wild patients - what does it mean. it is somewhat misleading. Perhaps the whole first sentence is not needed. I would recommend native English speaker to check the manuscript.

Some of the references are a bit dated - for example of Harvey. There is some more recent literature that can be used.

As per my comments above.

Author Response

Dear Editors and Reviewers
We appreciate your collaboration and support for our work.
The requested changes were mostly met and highlighted in the text. If necessary, we can make new corrections.
Yours sincerely.
Please see the attachment

Reviewer 2 Report

Dear Authors,

Congratulation on your good paper and research. The manuscript is clear and relevant but I think the structure can be improved. Several sentences included in "Materials and Methods" should be moved to "Results" and vice versa. In addition, the way the figures are proposed and the headings are written may confuse the reader. The conclusions are consistent with the arguments presented and are understandable in English. The work is interesting and original, but I have a few queries/change suggestions.

Lines 62-64: It would be better to cite the primary source when providing data.

Line 68: I think there is a problem with the bibliography. References must be numbered in order of appearance in the text. The problem repeats itself later, so I recommend a comprehensive revision of the bibliography.

Lines 73-76: I think this paragraph is not very fluent. I recommend changing the paragraph as follows, if this corresponds to the meaning the Authors want to express.

“Often oronasal fistulas do not cause clinical signs and it may be difficult to detect small fistulas during routine physical examination of unsedated and agitated patients, so anaesthesia may be required.”

Lines 77-80:  Since diagnosis is not the main topic of the study, I suggest deleting this paragraph and perhaps adding data on what has been published recently on surgical treatment. This would make the introduction more readable and link to the next paragraph on complications.

Lines 85-86: It might be interesting to add data on the recurrence rate in dogs.

Line 90: My greatest doubts concern the biocompatibility of this material. Its use is based solely on its binding to the enamel porosities and dentinal tubules. Are previous applications to the oral mucosa described?

Lines 99-100: Please replace “mucoperiostal” with “mucoperiosteal”.

Line 109: Please replace “were operated and evaluated” with “were included in this study”.

Line 113: Would it be correct to add 'based on surgical treatment performed'? This could immediately help the reader to understand the methods.

Lines 115-117: The Materials and Methods chapter should contain the methods and instruments used to conduct the study. In this case, the data obtained should be listed in the Results chapter. The error is repeated below in lines 119-120, 124-126 and 133-135. It would be more correct to state the inclusion criteria for each group studied in this section.

Lines 133-145: It may be unclear to the reader what the reason is for the classification into 3 groups: between G1 and G2 the difference is the surgical technique used (single or double flap); but what are the classification criteria for G3? Why are these groups used? What is the difference you want to highlight?

Figures 1 and 2: For the reader, a summary table of the cases included for each group could be very useful. Furthermore, you could include the reference number of the patient in question in the caption of each photo.

Figure 4: The photo is of very poor quality. I think it is better to offer the reader a collage highlighting the most representative clinical cases for each group at clinical presentation, after surgical treatment and during follow-up.

Lines 166-167: I think it is best to report the postoperative treatment after describing the surgical treatment carried out. It is also repeated in lines 186-187. I think it is better to combine the two sentences to avoid repetition.

Line 184: In my personal experience, this material does not adhere to the oral mucosa because it is designed to adhere to dentin and enamel. In the photos shown in the paper, the applied material seems to adhere to the oral mucosa only in the immediate post-operative phase. In later follow-up examinations, the self-curing glass ionomer cement adheres strongly to the sutures but not to the mucosa.

Line 193: How did you assess the healing of the surgical site? I think that the presence of the dressing did not allow a direct assessment of the wound. In addition, the results also consider a period after the patient has fully recovered. I would clarify in this section how often long-term follow-up were also performed to exclude recurrences.

Lines 201-202: In the G1 group, the flap and dressing were "performed an average of 15 days after the periodontal and extractive procedure that originated the defect," according to Materials and Methods. So how is it possible that "the removal of the cement protection occurred between 12 and 19 (15 ± 2) days after the maxillary canine tooth extraction"?

Lines 215-218: This "area of inflammation" that occurred in all patients could be due to the material used as a dressing. I think it is necessary to prove that this material can be used on the oral mucosa.

Figure 15 and 16: The photo are of poor quality and out of focus. As mentioned above, I believe that to facilitate reading, a summary table of the included cases for each group (G1, G2, and G3) should be added and each patient should be assigned a number, which is reproduced in the figures. This will make it easier for the reader to follow the patient through recovery.

Lines 237-239 and 244-245: The reader must take the trouble to recall the individual cases of the group under study. I point out again that it would facilitate the reading to assign a number (or other reference) to the patients.

Line 253: The nature of the "neoplastic mass" would be relevant to understanding healing and the possibility of recurrence.

Lines 268-269 and 272-273: Are a repetition of the introduction, please replace these sentences.

Line 292: I think "2013" is a typo.

Line 311: There is no "data" to compare, it is just a consideration of the authors. Were postoperative chest x-rays taken to rule out aspiration of fluid?

Lines 363-366: Did you forget to remove these sentences from the template?

Line 362: Some consideration of the materials used, compatibility with the oral mucosa, and previous applications in veterinary medicine should be included in the discussion.

Line 409: There is a more recent edition of this book.

Author Response

(The authors gave the same response as above.)

Reviewer 3 Report

Overall, this study provides good information and explains the benefits of glass ionomer cement in dental veterinary medicine very well. The reader can understand why the glass ionomer cement can be an alternative in the treatment of periodontal disease in the treatment of dogs. There are some points that could be edited or added:

1. Introduction: The current gold standard for the treatment of periodontal disease in dogs/animals should be highlighted and supported with studies. In addition, it should be described why the glass ionomer cement might have advantages. The aim of the study should be to be able to prove these advantages.

2. Materials and Methods: The methods should describe the follow-up interval. In which periods after treatment were the animals examined? What is your clinic standard?

3. In the discussion it should be mentioned why the treatment of periodontal disease should be carried out at all. Is there a guideline that describes exactly when the dog/animal needs surgical treatment?

4. Are there any other descriptions in the literature for covering the wound other than glass ionomer cement? What are the experiences and success rates here?

5. At the end of the discussions, the limitations of the study should be listed. For example, the number of animals in the groups is very small. Further studies with a larger number of animals should follow to verify the results of this study.

Author Response

(The authors gave the same response as above.)
